# Serum T2-High Inflammation Mediators in Eosinophilic COPD

**DOI:** 10.3390/biom14121648

**Published:** 2024-12-21

**Authors:** Andrius Januskevicius, Egle Vasyle, Airidas Rimkunas, Jolita Palacionyte, Virginija Kalinauskaite-Zukauske, Kestutis Malakauskas

**Affiliations:** 1Laboratory of Pulmonology, Department of Pulmonology, Lithuanian University of Health Sciences, LT-44307 Kaunas, Lithuania; egle.jurkeviciute@lsmu.lt (E.V.); airidas.rimkunas@lsmu.lt (A.R.); kestutis.malakauskas@lsmu.lt (K.M.); 2Institute of Biotechnology, Life Sciences Center, Vilnius University, LT-10257 Vilnius, Lithuania; 3Department of Pulmonology, Lithuanian University of Health Sciences, LT-44307 Kaunas, Lithuania; jolita.palacionyte@lsmu.lt (J.P.); virginija.kalinauskaite@lsmu.lt (V.K.-Z.)

**Keywords:** COPD, T2-high inflammation, eosinophil, mediators, biomarkers

## Abstract

Eosinophils are central inflammatory cells in asthma; however, a portion of patients with chronic obstructive pulmonary disease (COPD) have blood or sputum eosinophilia, a condition termed eosinophilic COPD (eCOPD), which may contribute to the progression of the disease. We hypothesize that eosinophilic inflammation in eCOPD patients is related to Type 2 (T2)-high inflammation seen in asthma and that serum mediators might help us to identify T2-high inflammation in patients and choose an appropriate personalized treatment strategy. Thus, we aimed to investigate ten serum levels of T2-high inflammation mediators in eCOPD patients and compare them to severe non-allergic eosinophilic asthma (SNEA) patients. We included 8 subjects with eCOPD, 10 with SNEA, and 11 healthy subjects (HS) as a control group. The concentrations of biomarkers in serum samples were analyzed using an enzyme-linked immunosorbent assay (ELISA). In this study, we found that eCOPD patients were distinguished from SNEA patients by elevated serum levels of sIL-5Rα, MET, TRX1, ICTP, and IL-4, as well as decreased serum levels of eotaxin-1 and sFcεRI. Moreover, MET, ICTP, eotaxin-1, and sFcεRI demonstrated high sensitivity and specificity as potential biomarkers for eCOPD patients. Furthermore, serum levels of IL-5 and IL-25 in combination with sIL-5Rα, MET, and IL-4 demonstrated a high value in identifying T2-high inflammation in eCOPD patients. In conclusion, this study highlights that while T2-high inflammation drives eosinophilic inflammation in both eCOPD and SNEA through similar mechanisms, the distinct expression of its mediators reflects an imbalance between T1 and T2 inflammation pathways in eCOPD patients. A combined analysis of serum mediators may aid in identifying T2-high inflammation in eCOPD patients and in selecting an appropriate personalized treatment strategy.

## 1. Introduction

Chronic obstructive pulmonary disease (COPD) and asthma affect millions of individuals around the world. The prevalence of COPD based on systematic reviews and meta-analysis might be around 480 million cases according to 2020 data [1,2], while the official numbers are lower. COPD is one of the leading causes of chronic illness and death globally and represents a major public health concern. COPD currently ranks fourth worldwide for top causes of death, with prevalence and mortality rates foreseen to continue to increase over time [3].

Asthma and COPD are distinct diseases, but they also share many mechanisms of action underlying them. These diseases differ in their symptoms, causes, course of the disease, inflammatory cells involved, types of airway inflammation, and mediators; however, a portion of COPD patients demonstrate similarities with severe asthma cases, with comparable airway inflammation and good responses to combined therapy, suggesting similar pathophysiologic characteristics [4]. Eosinophils are inflammatory cells that are common to COPD and asthma; based on different thresholds, approximately 10 to 40% of patients with COPD have eosinophilia in the blood or sputum [5,6,7,8], which may contribute to the progression of COPD and symptoms exacerbation [8,9,10]. Although both diseases share eosinophilic inflammation, their pathophysiology, disease progression, and response to therapy are very different. Understanding these differences enables clinicians to tailor treatments more effectively, such as the use of steroids or biological therapy. Moreover, correct differentiation will help to minimize the rates of exacerbation, reducing unnecessary side effects, and generally improve the quality of life, thus providing personalized care with regard to the peculiarities of each condition.

The recruitment and migration of eosinophils into the airways from the circulation initiate eosinophilic inflammation through processes influenced by chemokines’ C-C motif ligand, CCL11, and CCL5 [11]. This induces the turn upstream in cytokine secretion by epithelial cells in the airways in response to injurious stimuli and attracts T helper-2 cells and innate lymphoid cells into action, producing IL-5 that stimulates the maturation and release of eosinophils [12]. Eosinophils are increased both in the central and peripheral airways of patients with COPD, as shown by bronchial biopsies, bronchoalveolar lavage, or induced sputum [13]. These patients exhibit T2-high inflammation patterns similar to asthma patients, particularly those with high eosinophil counts. T2 inflammation-associated genes analysis revealed the association between eosinophils and IL-13, suggesting that these pathways may contribute to airway remodeling and mucus production in COPD, similar to mechanisms seen in asthma [14].

Treating a disease with standardized methods based only on diagnosis is less acceptable; therefore, a medical strategy based on the presence or absence of treatable traits is the aim of personalized therapy [15]. This highlights the importance of better understanding T2-endotype COPD with eosinophilic inflammation (eCOPD). In this study, we investigated serum levels of T2-high inflammation markers in eCOPD patients and compared them to SNEA patients. We chose well-known and less investigated analytes for understanding chronic inflammation conditions, blood oxidative stress markers, and collagen degradation levels. Moreover, to understand the pathogenetic relationship between eCOPD and severe asthma patients, we investigated eosinophils-related markers. The assessment of these bioactive substances could help to identify new biomarkers for eCOPD patients and aid in the development of targeted treatments; moreover, it would provide a better understanding of prevailing T2 inflammation.

## 2. Materials and Methods

### 2.1. Ethics Statement

All study participants were informed about the study and gave written informed consent. The study protocol was approved by the Regional Biomedical Research Ethics Committee of the Lithuanian University of Health Sciences (P3-BE-2-58/2020).

### 2.2. Study Design and Population

We recruited 8 COPD patients with a blood eosinophil count of at least 0.15 × 10^9^/L at screening or at least 0.3 × 10^9^/L during the previous year [16] and who had not used steroids for at least 1 year before the study, 10 SNEA patients who were using high doses of inhaled steroids, and 11 healthy subjects (HS) as controls. Group sizes were large enough for the determination of the statistical significance of serum protein expression, which was ensured by the robust sensitivity of the ELISA method, technical replicates, and the consistent expression patterns observed within the cohorts. Patients were from the Department of Pulmonology at the Hospital of the Lithuanian University of Health Sciences Kaunas Clinics. All participants were adults—men and women aged 18–80 years—who were informed about the study protocol and signed written informed consent.

Patients were included in the study by selecting them from electronic hospital medical records or when they arrived at the outpatient clinic for a consultation. All patients had to be approved according to the inclusion/exclusion criteria (Figure 1), and groups were homogenous according to the therapy of respiratory disease and had no clinically significant comorbidities. Within four weeks of their approval based on the inclusion and exclusion criteria, all eligible applicants were invited to participate in this study. Before the study began, each participant was given sufficient time to review, understand, and sign the informed consent form.

### 2.3. Pulmonary Function Tests

#### 2.3.1. Spirometry

Each participant completed spirometry testing a minimum of three times. Lung function was assessed using an ultrasonic spirometer (Ganshorn Medizin Electronic, Niederlauer, Germany). The procedure, along with detailed techniques, is outlined in [17]. Spirometry testing was performed according to updated 2022 ERS/ATS standards [18] for the interpretation of pulmonary function tests by using Global Lung Function Initiative normative equations.

#### 2.3.2. FeNO Test

FeNO analysis was performed for all study participants. A handheld Vivatmo-me device (Bosch Healthcare Solutions, Waiblingen, Germany) was used for the measurements, following the manufacturer’s instructions. The detailed methodology for this procedure is outlined in [19].

#### 2.3.3. Skin Prick Testing

All study participants underwent skin prick allergy testing for Dermatophagoides pteronyssinus, Dermatophagoides farinae, dog and cat dander, a mix of five grass pollens, birch pollen, mugwort, Alternaria, Aspergillus, and Cladosporium. Standardized allergen extracts (Stallergenes, S.A., Antony, France) were used for this procedure. The detailed methodology for performing this test is described in [17].

### 2.4. Blood Tests

#### 2.4.1. Complete Blood Count and Total IgE

Peripheral blood samples were collected from all participants into vacutainers containing dipotassium ethylenediaminetetraacetic acid (K2EDTA) (BD Vacutainer^®^; Becton Dickinson UK Ltd., Wokingham, UK). Routine clinical chemistry samples were promptly transported to the hospital laboratory. Complete blood count testing was conducted using the XE-5000 (Sysmex, Kobe, Japan) and UniCel^®^ DxH 800 Coulter^®^ Cellular Analysis System automated hematology analyzer (Beckman Coulter, Miami, FL, USA). Total IgE levels were measured using the AIA-2000 automated immunoassay analyzer (Tosoh Bioscience, South San Francisco, CA, USA).

#### 2.4.2. Measurement of Serum Levels of Selected Analytes by Enzyme-Linked Immunosorbent Assay

The serum levels of the circulating inflammatory markers soluble interleukin 5 receptor subunit alpha (sIL-5Rα), tyrosine-protein kinase (MET), interleukin (IL)-4, IL-5, IL-13, IL-25, eotaxin-1, and soluble high-affinity IgE receptor (sFcεRI), the blood oxidative stress marker thioredoxin 1 (TRX1), and the collagen degradation product C-telopeptide of type I collagen (ICTP) were measured using an enzyme-linked immunosorbent assay according to the manufacturer’s instruction. To control the batch effect and technical variations, every sample was investigated using three replicates in the ELISA plate, and identified outliers were eliminated from the analysis. For all analytes, a total of two ELISA plates were ordered from the same batch—SNEA and eCOPD samples were analyzed in the same plate, while HS samples were analyzed in separate plates. The distribution of standard curve values in both plates was used to identify the quality of reproducibility. The sensitivity for ELISA kits was as follows: IL-4, 1.3 pg/mL; IL-5, 1.45 pg/mL; IL-13, 0.7 pg/mL; IL-25, 0.4 ng/mL; sIL-5Rα, 150 pg/mL; eotaxin-1, 2.2 pg/mL; sFcεRI, 68 pg/mL; ICTP 50 pg/mL; TRX1, 0.2 ng/ mL; MET, 14.65 pg/mL. Due to technical limitations, some concentrations of biologically active substances fell below the sensitivity range and were therefore recorded as zero values.

### 2.5. Statistical Analysis

Statistical analysis was calculated with GraphPad Prism 8 for Windows (ver. 8.01, 2018; GraphPad Software Inc., San Diego, CA, USA). Both descriptive and analytical statistical methods were applied. To assess the normality of data distribution, the Shapiro–Wilk test was used. Since the data did not meet the assumption of normality, and the sample sizes in each group were small, the nonparametric Mann–Whitney U-test (two-sided) was employed to compare two independent groups. Spearman’s rank correlation coefficient was used to evaluate correlations between datasets. The sensitivity and specificity of the indicators were assessed using receiver operating characteristic (ROC) curve analysis. A *p*-value of less than 0.05 was considered statistically significant.

## 3. Results

### 3.1. Study Subject Characteristics

A total of 29 investigated individuals aged 18 to 81 were included in the study (8 eCOPD patients, 10 SNEA patients, and 11 HSs). Table 1 summarizes the demographic and clinical data of the studied groups. eCOPD and SNEA patients were significantly older compared to the HS; moreover, the sample of eCOPD patients included more men than females, while the SNEA patients and HS were homogenous in this criterion (Table 1). All investigated individuals had negative skin prick tests. eCOPD patients were current or ex-smokers, while those in the SNEA and HS groups were non-smokers.

The highest blood EOS count was found in the SNEA group, while eCOPD patients’ blood EOS count was lower than that of SNEA patients, but was significantly higher compared to the HS (*p* < 0.05). There were no differences between the diseased groups in FEV_1_ (L); however, the eCOPD was distinguished from all other groups by FEV_1_, expressed as a percentage. Significantly lower Fe_NO_ levels were observed in the eCOPD and HS groups compared to the SNEA group. Blood total IgE levels were increased in the diseased groups compared to the HS (*p* < 0.05, Table 1).

### 3.2. Serum Levels of Chronic Inflammation, Blood Oxidative Stress, and Collagen Degradation Markers in the Eosinophilic Type of Chronic Lung Diseases

We chose to compare the concentrations of the inflammatory markers sIL-5Rα and MET, the oxidative stress marker TRX1, and the collagen degradation marker ICTP in the investigated subject groups. In the eCOPD group, sIL-5Rα levels were elevated compared to the severe asthma group but similar to the HS (*p* < 0.05). sIL-5Rα levels in the SNEA group were found to be lower compared to the other groups (*p* < 0.05) (Table 2, Figure 2A).

MET serum concentration was found to be enhanced in eCOPD patients, being higher than in the SNEA and HS groups (*p* < 0.05). Based on the ROC curve analysis, the AUC was 0.97 (*p* < 0.001) and 0.79 (*p* < 0.05), respectively (Figure 3A,B). MET levels in the SNEA group were not significantly different compared to the HS (*p* > 0.05).

Serum blood oxidative stress marker TRX1 levels show a distinct distribution among the investigated groups. In the eCOPD group, TRX1 levels were enhanced over the SNEA group (*p* < 0.05) but were comparable with the HS (*p* > 0.05). The SNEA group had decreased serum levels of TRX1 compared to the other groups (*p* < 0.05) (Table 2, Figure 2C).

Lastly, concentrations of the collagen degradation marker ICTP in the blood were the highest in the eCOPD patients and significantly differed from other groups (*p* < 0.05)—based on the ROC curves analysis, the AUC was 0.96 (*p* < 0.001) and 1.00 (*p* < 0.05), respectively (Figure 3B,C). Moreover, while ICTP levels in the SNEA group were lower compared to eCOPD group, they were enhanced compared to the HS (*p* < 0.05) (Table 2, Figure 2D).

### 3.3. Heterogeneity in Serum Levels of T2-High Inflammation Mediators in eCOPD and Severe Eosinophilic Asthma Patients

We chose to analyze well-documented T2-high mediators indicated to be increased in asthma compared to healthy controls; thus, we do not include the HS group in the second part of this study. eCOPD patients could be distinguished from the severe asthma group by enhanced IL-4 levels (Table 3, Figure 4A) and reduced serum levels of eotaxin-1 and sFcεRI (*p* < 0.05) (Table 3, Figure 4E,F). Based on the ROC curves analysis, the AUC of eotaxin-1 for eCOPD vs. SNEA was 1.00 (*p* < 0.001) (Figure 5A), while for sFcεRI for eCOPD vs. SNEA, it was 0.87 (*p* < 0.01) (Figure 5B). There were no significant differences in serum levels of IL-5, IL-13, and IL-25 among the investigated groups (*p* > 0.05) (Table 3, Figure 4B–D).

### 3.4. Correlations Between Serum Levels of T2-High Inflammation Mediators in eCOPD Patients

We evaluate the relationship between the analyzed serum biomarkers in eCOPD patients to better understand their co-regulation and potential prognostic and diagnostic value. Spearman’s correlation analysis reveals a positive correlation between ICTP and eotaxin-1 (Table 4, Figure 6A). sIL-5Rα demonstrates a positive correlation with MET, IL-5, and IL-25 (Table 4, Figure 6B–D); furthermore, IL-5 correlates with IL-25, MET, and IL-4 as well (Table 4, Figure 6F,H,I). In eCOPD patients, the serum levels of IL-25 show a positive correlation not only with sIL-5Rα and IL-5, but also with IL-4 and MET (Table 4, Figure 6E,G). Finally, the analysis revealed that FcεRI shows a negative correlation with the serum levels of eotaxin-1 (Table 4, Figure 6J.

## 4. Discussion

Our study aimed to investigate the serum levels of T2-high inflammation mediators in T2-endotype COPD and compare them with SNEA patients. We found that eCOPD patients were distinguished from SNEA patients by elevated serum levels of sIL-5Rα, MET, TRX1, ICTP, and IL-4 and decreased serum levels of eotaxin-1 and sFcεRI. MET, ICTP, eotaxin-1, and sFcεRI exhibit high sensitivity and specificity, making them promising biomarkers for identifying T2-endotype COPD patients. Furthermore, the combined assessment of serum IL-5 and IL-25, along with sIL-5Rα, MET, and IL-4, shows significant potential in identifying T2-high inflammation in eCOPD patients.

COPD is usually described as a heterogeneous disease characterized by persistent respiratory symptoms and airflow limitation. However, it is more commonly known for respiratory symptoms such as progressive dyspnea, chronic cough, sputum production, chest tightness, or fatigue [20,21], while asthma receives more attention regarding the predominant airway inflammation [22]. This narrative probably exists due to the fact that the main focus in treating COPD is the symptomatic treatment of bronchial obstructions, rather than the anti-inflammatory therapy seen in asthma [23]; however, airway inflammation in COPD is related to disease mortality and progression [24]. While neutrophil-associated COPD is the most prevalent phenotype, a portion of these patients demonstrate an increased eosinophilic inflammation in the blood or sputum [7,11], and can be called eosinophilic, eCOPD, or T2-endotype COPD. Similarly to asthma, there is an association between eosinophilic inflammation and a higher risk of future severe exacerbations [25,26]. Innate and adaptive immunity crosstalk in asthma pathogenesis is well described; however, in eCOPD patients, it remains uncertain. Eosinophilic inflammation may also arise from ILC2 cells producing IL-5 and IL-13 in response to epithelial alarmins IL-25, IL-33, and TSLP; however, the current mechanisms driving eosinophilic inflammation are unclear. While the ILC2 pathway is often considered more pivotal than the Th2 pathway, as eosinophilic COPD is frequently associated with non-allergic inflammation, environmental irritants, and infections rather than allergen-driven responses, Th2 pathways may still contribute significantly. The secretion of IL-4, IL-5, and IL-13 by T-helper cells promotes the survival of eosinophils [27], while IL-4 and IL-13 affect IgE production.

Blood and sputum eosinophil numbers are the main characteristics used for the identification of T2-endotypes; however, there is still a lack of consensus regarding the current value thresholds. A lot of T2 inflammation markers can be identified and are easily detectable in the serum of asthma patients and might help in determining the level of T2 inflammation in eCOPD patients in order to choose an appropriate treatment strategy. At first, we determined the levels of the inflammatory markers sIL-5Rα and MET, the oxidative stress marker TRX1, and the collagen degradation marker ICTP in the groups. sIL-5Rα emerges through either alternative splicing of the mRNA transcript or proteolytic cleavage of the membrane-bound receptor. By releasing sIL-5Rα into the bloodstream, the body can regulate IL-5 activity as it can bind to circulating IL-5, potentially reducing its ability to interact with membrane-bound receptors and therefore modulating the eosinophilic response [28,29,30]. sIL-5Rα levels in eCOPD patients did not differ from the HS but were almost ten times higher than in the SNEA patients (Figure 2A). The distinct sIL-5Rα levels in eCOPD and severe asthma patients suggest differentially regulated IL-5 signaling pathways. Proteolytic cleavage of IL-5Rα or its alternative splicing occurs to diminish receptor numbers on expressing cells or to lower the circulating IL-5 levels after IL-5/sIL-5Rα complex formation. While circulating IL-5 levels are enhanced in SNEA patients compared to HS [31], its activity regulation via the production of sIL-5Rα does not occur, and serum levels of sIL-5Rα remain decreased (Figure 2A). Furthermore, as IL-5 levels were equal between diseased groups (Figure 5B), we can assume that in eCOPD patients, it is enhanced over HS as well. However, sIL-5Rα levels are not higher than in HS, as seen in SNEA patients, indicating that the IL-5 signaling pathway’s activity in eCOPD patients is partly downregulated. Targeting T2-mediated inflammation using biological therapies dramatically improved clinical practice in severe asthma. However, anti-IL5 or anti-IL-5R does not achieve appropriate success for eCOPD patients. Both benralizumab and mepolizumab treatment reduced eosinophilic inflammation, but the primary outcome of reducing the annual rate of acute exacerbations was not met or reduction was low [16,32,33]. Higher sIL-5Rα levels in eCOPD patients compared to severe asthma patients indicate enhanced receptor cleavage of cell membranes and the neutralization of IL-5 in patients’ blood. This might affect biological medication’s action, as organisms are adapted to regulate IL-5 activity.

Similarly to IL-5, MET signaling is closely associated with immune cell activation and the production of cytokines and chemokines involved in disease pathogenesis. MET activation can boost the recruitment and activation of inflammatory cells in the airways, like eosinophils and T lymphocytes. Additionally, the MET-triggered release of cytokines, such as interleukin-8 and interleukin-13, can intensify airway inflammation and worsen disease severity [34,35,36]. However, its current role in COPD patients has not been explored. The enhanced MET serum levels in eCOPD patients compared to other groups raises interest in ectodomain shedding from mainly epithelial and endothelial cells that need deeper investigation. MET is a receptor for hepatocyte growth factor, and this signaling axis is responsible for cell growth, regeneration, and wound healing processes [37]. While the downregulation of MET signaling is a target for lung cancer [38], enhanced shedding leading to dysregulated signaling might promote epithelium remodeling in eCOPD patients. Moreover, this might stimulate investigations into the tissue inhibitors of metalloproteinases deeper in eCOPD as they control the shedding process of MET ectodomain [39]. Moreover, ROC curve analysis revealed its diagnostic value over the SNEA and HS groups (Figure 3A,B), and a significant positive correlation of MET and sIL-5Rα in eCOPD patients (Figure 6B) suggests its value for T2-high inflammation identification and understanding of its mechanisms and the application of appropriate treatment.

Among the known culprits that trigger and enhance the risk of COPD are smoking and air pollution. These factors significantly enhance oxidative stress—the major cause of cellular damage to the airways. The lungs are particularly susceptible to this type of stress because they are always exposed to oxygen, have a rich supply of blood, and come into contact with many different pathogens and toxins within the environment [40]. Proteins involved in oxidative stress regulation such as TRX1 might be used as biomarkers to assess oxidative stress levels and monitor the progression of the disease. TRX1 is a protein that plays an essential role in cellular defense mechanisms against oxidative stress by helping to neutralize reactive oxygen that is abundant in COPD due to cigarette smoke, pollution, and other environmental factors. It acts as a redox regulator, antioxidant, anti-allergic, and anti-inflammatory molecule [41,42]. TRX1 levels were higher in eCOPD patients’ serum compared to those with severe asthma; however, it did not differ from the HS (Figure 2C). The normal levels of TRX1 in eCOPD patients and decreased levels in the SNEA group require further clarification. This risk factor indicates that higher oxidative stress in organisms, rather than the disease itself, enhances the production of the antioxidant system components. The eCOPD patients are all current or former smokers, unlike the asthma patients in this study. Smoking exposes the lungs to significant particulate matter, prompting an initial T1 immune response aimed at clearing debris, driven by M1 macrophages and dendritic cells. However, in heavy smokers, this response is often suppressed, increasing susceptibility to infections. Chronic exposure to high particulate levels can shift the balance toward M2/T2 phenotypes as T1 responses wane. This creates a continuum of T1 and T2 responses in smokers, influenced by smoking intensity and duration. The levels of TRX1 could therefore be partly explained by this status, as smokers have significantly higher serum thioredoxin concentrations [43]. Moreover, TRX1 acts not only as a regulator of cellular redox homeostasis. It regulates the T1/T2 immune balance [44] and affects eosinophils’ [45,46] and neutrophils’ [47,48] activity and migration. Enhanced eosinophilic inflammation in SNEA patients might be related to decreased TRX1 levels as eosinophil infiltration is not suppressed; moreover, steroid treatment was found to decrease serum levels of TRX1 [49]. Based on these data, TRX1 concentration in eCOPD patient blood should have decreased as well; however, long-term adaptations may allow TRX1 levels to be maintained within the normal range as part of a balanced compensatory mechanism.

There is increased extracellular matrix (ECM) turnover in COPD with enhanced collagen I degradation; moreover, this is related to disease severity, including airflow limitation, dyspnea, and time to exacerbation [50]. ICTP is a collagen I degradation fragment that can be detected in the serum and serves as a marker of airway remodeling [51]. The analysis of serum ICTP levels provides valuable insights into collagen degradation in patients—concentrations were the highest in the eCOPD, showing statistically significant differences from other groups (Figure 2D) with high specificity and sensitivity values (Figure 3C,D). This elevation in ICTP underscores the pronounced collagen degradation and ECM remodeling, likely due to heightened inflammatory responses and increased proteolytic activity. Further, while the highest serum ICTP levels were found in the eCOPD, SNEA patients had elevated ICTP concentrations compared to the control as well. This highlights the importance of ECM remodeling during obstructive diseases and indicates that collagen degradation fragments might be biomarkers for eCOPD and SNEA.

Compared to asthma, the T2 pathways in eCOPD are less well documented and may differ, but they still share significant similarities primarily with severe non-allergic asthma. However, our findings show heterogeneity in blood T2-high inflammation mediators between the investigated groups. We chose to analyze well-documented T2-high mediators indicated to be increased in asthma compared to healthy controls; thus, we do not include the HS group in the second part of the manuscript, and instead compared eCOPD patients with severe asthma patients. Activated T2 cells secrete cytokines, notably interleukin-4 (IL-4), interleukin-5 (IL-5), and interleukin-13 (IL-13). IL-5 is crucial for the growth, differentiation, and survival of eosinophils. IL-4 and IL-13 induce the expression of adhesion molecules on endothelial cells, facilitating eosinophil migration from the bloodstream into lung tissues. Additionally, these cytokines stimulate the production of eotaxins (CCL11, CCL24, and CCL26), chemokines that create a gradient guiding eosinophils to sites of inflammation [52]. Increased IL-4 and IL-13 levels over healthy controls are a hallmark of T2-high asthma and are important in eCOPD as well [11,53]. However, our findings demonstrate that eCOPD patients had enhanced serum concentrations of IL-4 compared to SNEA patients; however, no differences between these groups in IL-13 levels were found, although a tendency for the IL-13 levels to be enhanced in eCOPD patients can be observed (Figure 4A,C). There are no current explanations for why the IL-4 and IL-13 signaling axis might be more pronounced in eCOPD patients. Activated ILC2 and Th2-cells are essential for the production of IL-4 and IL-13, which subsequently promote B cells to produce IgE [54,55]. This partly explains why IgE levels are enhanced in eCOPD patients while the patient population was non-atopic (Table 1). While SNEA is non-atopic by definition, eCOPD involves cytokines typically associated with sensitization, such as IL-4 and IL-13, which might contribute to IgE production through non-classical pathways. This difference could reflect a distinct immunological profile, highlighting the heterogeneity in its inflammatory mechanisms compared to SNEA.

Other selected T2-high inflammation mediators related to eosinophils directly, IL-25 and eotaxin-1, are both critical players in the regulation of eosinophil recruitment and activation conditions characterized by eosinophilic inflammation such as asthma and eCOPD [9,56]. However, the function of IL-25 in eCOPD patients is not entirely clear, as elevated levels of IL-25 in blood plasma have been linked to favorable outcomes, showing an association with a lower risk of future moderate-to-severe exacerbations [57]. This is the opposite effect to that seen in asthma, where high IL-25 levels are related to greater airway hyperresponsiveness, more airway and blood eosinophils, more subepithelial thickening, and the higher expression of T2-high signature genes [58]. This might be explained by the imbalance between T1 and T2 responses which might be both dominant in eCOPD patients, as well as the independent functions and multifactorial regulation of eosinophils and IL-25. There is an increasing realization of the potential overlap between T2 and T1 inflammatory mechanisms in driving COPD. The contributions of these different pathways to pathological changes, and the links between inflammation, pathology, and clinical characteristics, are under active investigation. While there is some overlap between T1 and T2 inflammatory processes, the particular features of each would make the manifestations different in the presentation of the disease, its clinical features, and its management.

While IL-25 serum levels in eCOPD patients were as high as in severe asthma (Figure 4D), eotaxin-1 levels differed between groups. Eotaxin-1 levels were lower in eCOPD patients compared to SNEA patients (Figure 4E). Moreover, ROC curve analysis demonstrates that eotaxin-1 is a significant marker in distinguishing eCOPD (Figure 5A). Eotaxin, primarily produced by epithelial cells, activated fibroblasts, and endothelial cells, is a potent eosinophil-specific chemokine that selectively attracts eosinophils to sites of inflammation [59]. It is not clear why in both diseases with predominant eosinophilic inflammation eotaxin levels are different. Blood and airway eosinophilia is enhanced in eCOPD patients compared to HS, but it is not as intense as in patients with severe asthma. This might be related to lower eotaxin-1 levels led by less activated eotaxin-producing cells—a consequence of less pronounced T2-high inflammation. Eosinophil recruitment by eotaxin-1 is regulated via CC chemokine receptor 3 (CCR3) [60]; however, eCOPD patients have notably enhanced neutrophil numbers as well. CCR3 is not a receptor normally expressed on neutrophils, but during chronic inflammatory conditions, they acquire novel chemokine receptor expression [61]. Lower eotaxin-1 levels during eCOPD compared to SNEA might be explained in part due to consumption through receptor binding influencing neutrophil migration as well. Moreover, eosinophil recruitment is regulated by many different mechanisms, and we can assume that eotaxin-1 might not be playing a main role in attracting eosinophils to the airways. This is in contrast to asthma, where eotaxin is a key player in directing eosinophil migration.

We also examined serum levels of the soluble receptor FcεRI, a high-affinity IgE receptor expressed in various cell types, including eosinophils [62,63]. IgE is mostly related to atopy; however, in our study, higher levels of sFcεRI were found in the SNEA group (Figure 4F). Moreover, sFcεRI shows a very high value as a prognostic marker to distinguish eCOPD (Figure 5B). In contrast, the highest IgE levels were found in the eCOPD group, and data exist that show that IgE regulates the expression of high-affinity receptors on the cell surface [64,65]; however, how this might be related to its soluble forms in the bloodstream remains unknown. In the bloodstream, sFcεRI typically exists in two main forms: a free form and a complex form bound to IgE. sFcεRI enters the bloodstream primarily through proteolytic cleavage or shedding from the cell surface and after alternative splicing of the FcεRI mRNA [66,67]. Higher IgE levels and the ability of the free form of sFcεRI to interact with IgE might provide an explanation for why its levels are lower than in severe asthma; moreover, sFcεRI can interfere with IgE detection in the serum, which might be of importance regarding interference in IgE detection and diagnosis [68]. Moreover, non-allergic eCOPD patients are less dependent on the IgE–FcεRI axis, and lower expression of FcεRI in inflammatory cells leads to lower levels of its soluble forms in the blood after proteolytic cleavage and/or shedding. However, this does not explain why serum levels of sFcεRI are higher in SNEA patients but decreased in non-allergic eCOPD patients. Cigarette smokers are reported to display decreased FcεRI expression on mast cells [69], and this might affect other inflammatory cells as well. However, such differences in sFcεRI levels and other mediators highlight the fact that these diseases are similar, but also very different in the activation of pathogenetic pathways, where the relationship between the T1 and T2 pathways is a critical factor at play and needs to be better investigated.

A combined analysis of T2 biomarkers in eCOPD patients might help to better distinguish T2-endotype COPD patients and treat them with a strategy of personalized therapy. The T2-high inflammation is intricately linked to a web of cytokines and chemokines that communicate extensively to regulate immune responses. This cross-communication is essential, as it orchestrates both pro-inflammatory and anti-inflammatory signals, influencing disease progression and potentially guiding therapeutic targets [70]. Our study provides novel insights into the complex cytokine and chemokine network within eCOPD by evaluating the relationships among specific serum biomarkers. Using Spearman’s correlation analysis, we identified that sIL-5Rα shows strong correlations with MET, IL-5, and IL-25, while IL-5 correlates with MET, IL-4, and IL-25, indicating coordinated signaling pathways that might contribute to eosinophil recruitment and T2 amplification in eCOPD. In particular, IL-25’s positive correlation with sIL-5Rα, IL-5, IL-4, and MET further underscores its role as a central mediator in eCOPD-related T2 inflammation. Interestingly, FcεRI showed a unique negative correlation with eotaxin levels, which could imply a regulatory counterbalance that limits excessive eosinophil activation (Table 4, Figure 6).

This study had several limitations. To better understand the inflammatory markers’ diagnostic and prognostic value, a large-scale study with highly investigated individual populations is mandatory. Moreover, data in the SNEA group might be affected by steroid use, due to their anti-inflammatory effect, which might reduce marker expression, while the eCOPD group consisted of steroid-free patients. However, we considered that SNEA patients showed resistance to steroid treatment, and anti-IL-5 therapy was included in their treatment plan, allowing us to assume that the effect of steroids was minor. Smoking status might affect the expression of inflammation mediators in eCOPD patients; however, research in the context of smoking is an integral part of COPD. Additional details about serum levels of T2-high inflammation mediators in current smokers and ex-smokers groups are essential. Finally, the inclusion of non-eosinophilic COPD patients in the study would have provided supplementary interpretation about T2-high inflammation activity in eCOPD patients.

## 5. Conclusions

Our study contributes to the growing body of evidence on cytokine interplay in eCOPD, offering potential biomarkers for prognosis and treatment targeting in T2-driven COPD. This study emphasizes the different T2-high inflammation mediator profiles across eCOPD and severe asthma, suggesting them as valuable indicators in distinguishing these disease entities. Moreover, this study highlights that while T2-high inflammation operates through similar mediators enhancing eosinophilic inflammation in both diseases, its distinct expression levels demonstrate the imbalance between T1 and T2 inflammatory pathways, where both contribute to the inflammatory process in eCOPD patients. However, a comprehensive analysis of serum mediators could assist in detecting T2-high inflammation in eCOPD patients and guide the selection of personalized treatment strategies.

## Figures and Tables

**Figure 1 biomolecules-14-01648-f001:**
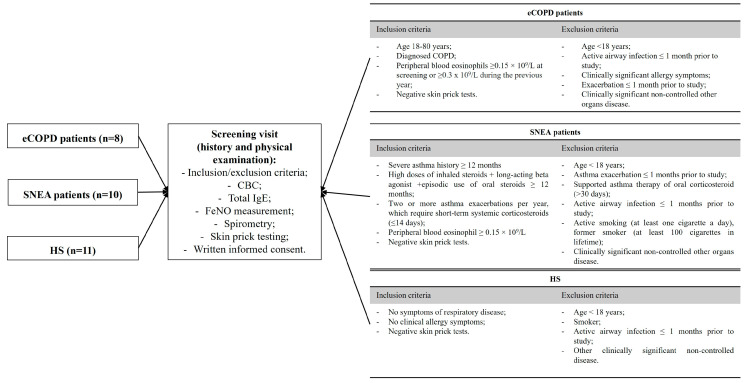
Inclusion and exclusion criteria of the study population. CBC—complete blood count; FeNO—fractional exhaled nitric oxide; IgE—immunoglobulin E; eCOPD—eosinophilic chronic obstructive pulmonary disease; SNEA—severe non-allergic eosinophilic asthma; HS—healthy subjects.

**Figure 2 biomolecules-14-01648-f002:**
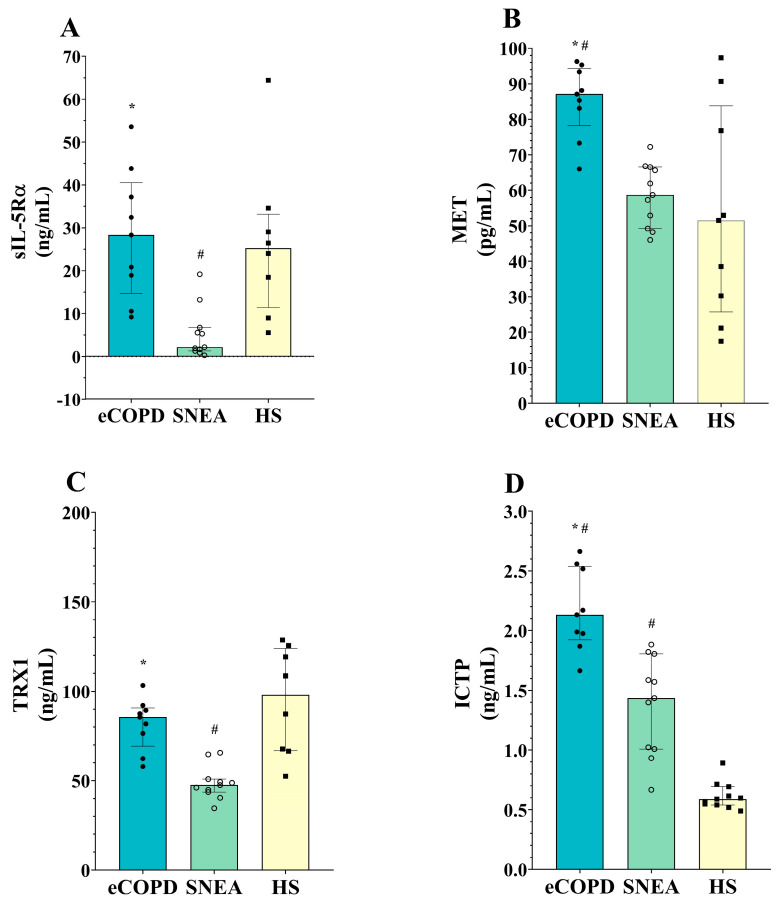
Serum levels of the inflammatory mediators sIL-5Rα (**A**) and MET (**B**), the oxidative stress marker TRX1 (**C**), and the collagen degradation marker ICTP (**D**). eCOPD—eosinophilic chronic obstructive pulmonary disease; SNEA—severe non-allergic asthma; HS—healthy subjects; TRX1—thioredoxin 1; ICTP—C-telopeptide of type I collagen; MET—tyrosine-protein kinase Met; sIL-5Rα—soluble interleukin 5 receptor subunit alpha. Data are presented as the median with the interquartile range. Statistical analysis: between investigated, Mann–Whitney two-sided U-test. * *p* < 0.05 compared to SNEA; ^#^ *p* < 0.05 compared to HS.

**Figure 3 biomolecules-14-01648-f003:**
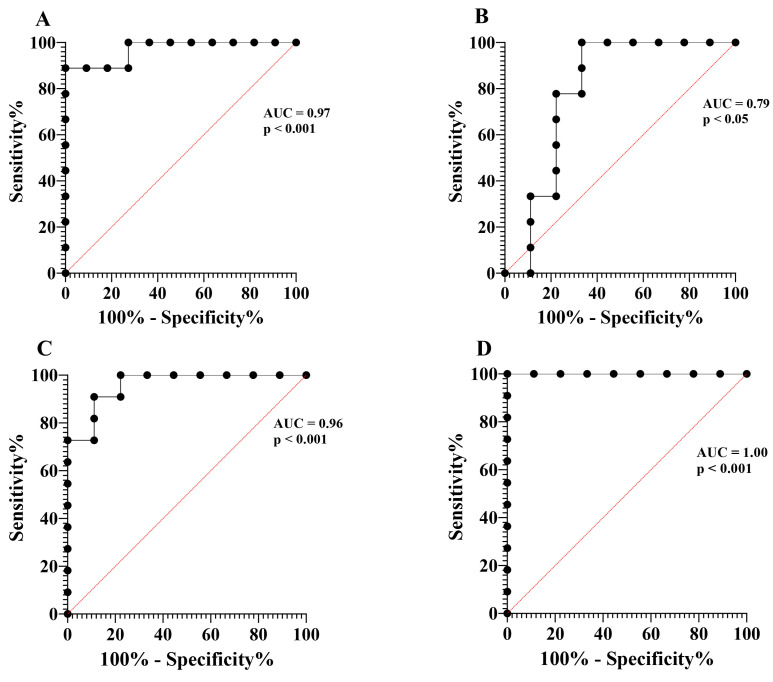
ROC curves of MET and ICTP. ROC curve of MET—eCOPD vs. SNEA (**A**); ROC curve of MET—eCOPD vs. HS (**B**); ROC curve of ICTP—eCOPD vs. SNEA (**C**); ROC curve of ICTP—eCOPD vs. HS (**D**). AUC—area under the curve.

**Figure 4 biomolecules-14-01648-f004:**
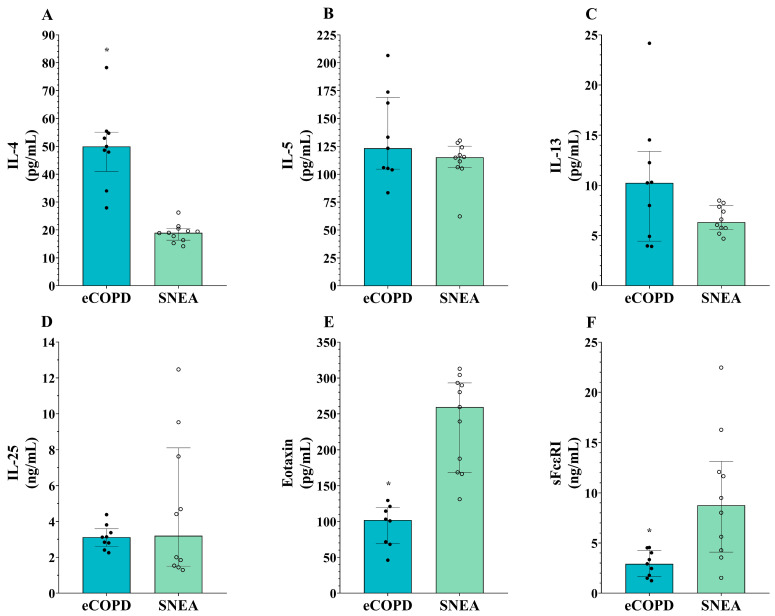
Serum levels of T2-high inflammation mediators IL-4 (**A**), IL-5 (**B**), IL-13 (**C**), IL-25 (**D**), eotaxin-1 (**E**), and sFcεRI (**F**). eCOPD—eosinophilic chronic obstructive pulmonary disease; SNEA—severe non-allergic asthma; IL—interleukin; sFcεRI—soluble high-affinity IgE receptor. Data are presented as the median with an interquartile range. Statistical analysis: between investigated groups, Mann–Whitney two-sided U-test. * *p* < 0.05 compared to the SNEA.

**Figure 5 biomolecules-14-01648-f005:**
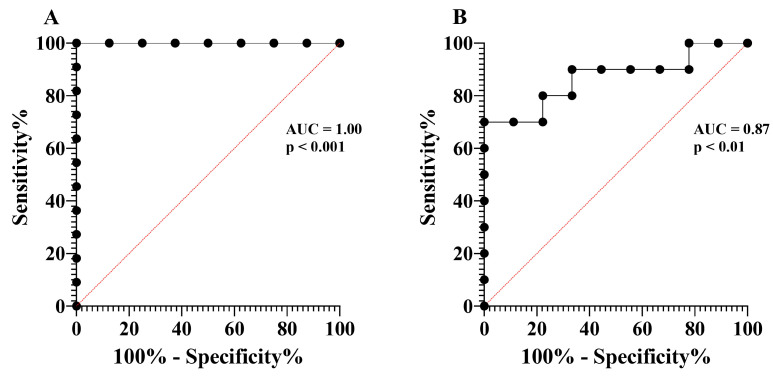
ROC curves of eotaxin-1 and sFcεRI. ROC curve of eotaxin-1—eCOPD vs. SNEA (**A**); ROC curve of sFcεRI—eCOPD vs. SNEA (**B**). AUC—area under the curve.

**Figure 6 biomolecules-14-01648-f006:**
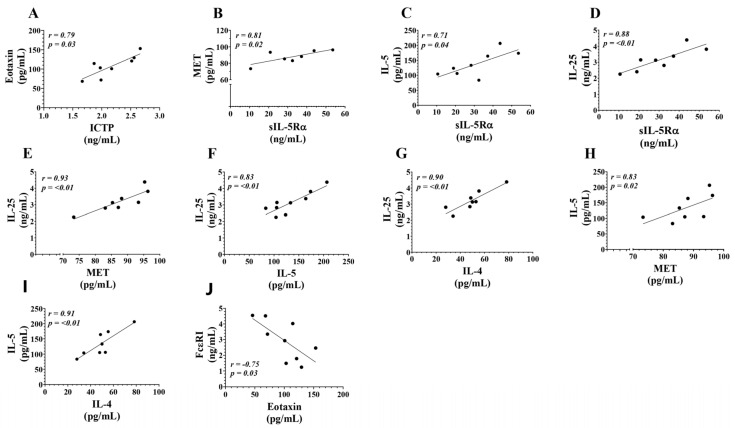
Correlations between serum levels of T2-high mediators in eCOPD patients. ICTP and eotaxin-1 (**A**); sIL-5Rα and MET (**B**); sIL-5Rα and IL-5 (**C**); sIL-5Rα and IL-25 (**D**); IL-25 and MET (**E**); IL-25 and IL-5 (**F**); IL-25 and IL-4 (**G**); IL-5 and MET (**H**); IL-5 and IL-4 (**I**); FcεRI and eotaxin-1 (**J**). TRX1—thioredoxin 1; ICTP—C-telopeptide of type I collagen; MET—tyrosine-protein kinase Met; sIL-5Rα—soluble interleukin 5 receptor subunit alpha; IL—interleukin; sFcεRI—soluble high-affinity IgE receptor; r—correlation coefficient. Statistical analysis: Spearman’s rank correlation.

**Table 1 biomolecules-14-01648-t001:** Demographic and clinical characteristics of the study population.

Characteristic	eCOPD	SNEA	HS
**Number, n**	8	10	11
**Sex, M/F**	6/2	4/6	4/7
**Age, years**	61.3 ± 3.4 ^#^	62.7 ± 1.0 ^#^	31.5 ± 2.8
**BMI, kg/m^2^**	30.9 ± 1.4 ^#^	27.4 ± 2.0	25.0 ± 1.6
**FEV_1_, L**	1.8 ± 0.2 ^#^	1.8 ± 0.2 ^#^	3.8 ± 0.3
**FEV_1_, % of predicted**	47.9 ± 4.6 *^#^	70.4 ± 8.2 ^#^	99.6 ± 6.3
**Fe_NO_, ppb**	15.0 ± 4.1 *	34.3 ± 6.5 ^#^	18.6 ± 6.2
**Blood EOS count, × 10^9^/L**	0.27 ± 0.02 *^#^	0.71 ± 0.16 ^#^	0.15 ± 0.04
**Total IgE, IU/mL**	101.8 ± 41.54 ^#^	168.3 ± 39.5 ^#^	16.7 ± 4.6

BMI—body mass index; EOS—eosinophil; F—female; Fe_NO_—fractional exhaled nitric oxide; FEV_1_—forced expiratory volume in 1 s; IgE—immunoglobulin E; M—male; eCOPD—eosinophilic chronic obstructive pulmonary disease; SNEA—severe non-allergic asthma; HS—healthy subjects. Data presented as the mean ± standard error of the mean. Statistical analysis: between investigated groups, Mann–Whitney two-sided U-test. * *p* < 0.05 compared to SNEA; ^#^ *p* < 0.05 compared to HS.

**Table 2 biomolecules-14-01648-t002:** The concentration of selected chronic inflammation, blood oxidative stress and collagen degradation markersin the investigated groups.

	eCOPD	SNEA	HS
**sIL-5Rα (ng/mL)**	28.3 ***** [14.7; 40.5]	2.2 ^#^ [1.3; 6.7]	25.2 [11.3; 33.2]
**MET (pg/mL)**	87.1 *^#^ [78.2; 94.3]	58.7 [49.2; 66.5]	51.5 [25.7; 83.8]
**TRX1** **(ng/mL)**	85.5 * [69.4; 70.7]	47.5 ^#^ [43.7; 50.8]	98.0 [66.8; 123.9]
**ICTP** **(ng/mL)**	2.1 *^#^ [1.9; 2.5]	1.4 ^#^ [1.0; 1.8]	0.59 [0.54; 0.69]

eCOPD—eosinophilic chronic obstructive pulmonary disease; SNEA—severe non-allergic asthma; HS—healthy subjects; TRX1—thioredoxin 1; ICTP—C-telopeptide of type I collagen; MET—tyrosine-protein kinase Met; sIL-5Rα—soluble interleukin 5 receptor subunit alpha. Data are presented as the median with the interquartile range. Statistical analysis: between investigated groups, Mann–Whitney two-sided U-test. * *p* < 0.05 compared to SNEA; ^#^ *p* < 0.05 compared to HS.

**Table 3 biomolecules-14-01648-t003:** The concentration of selected T2-high inflammation mediators in the investigated groups.

	eCOPD	SNEA
**IL-4 (pg/mL)**	50.0 * [41.0; 55.1]	19.0 [16.4; 20.5]
**IL-5 (pg/mL)**	123.4 [104.6; 168.8]	115.1 [106.0; 125.3]
**IL-13 (pg/mL)**	10.3 [4.4; 13.4]	6.3 [5.6; 8.5]
**IL-25 (ng/mL)**	3.1 [2.6; 3.6]	3.2 [1.5; 8.1]
**Eotaxin-1 (pg/mL)**	103.2 * [69.8; 125.1]	259.4 [168.6; 293.1]
**sFcεRI (ng/mL)**	2.9 * [1.6; 4.3]	8.8 [4.1; 13.1]

eCOPD—eosinophilic chronic obstructive pulmonary disease; SNEA—severe non-allergic asthma; IL—interleukin; sFcεRI—soluble high-affinity IgE receptor. Data are presented as the median with an interquartile range. Statistical analysis: between investigated groups, Mann–Whitney two-sided U-test. * *p* < 0.05 compared to the SNEA.

**Table 4 biomolecules-14-01648-t004:** Correlation of serum T2-high inflammation mediators of eCOPD patients.

	ICTP	TRX1	sIL-5Rα	MET	IL-4	IL-5	IL-13	IL-25	Eotaxin-1	FcεRI
**ICTP**		r = 0.28 (*p* = 0.46)	r = −0.03 (*p* = 0.95)	r = −0.20 (*p* = 0.61)	r = 0.40 (*p* = 0.29)	r = 0.17 (*p* = 0.68)	r = 0.22 (*p* = 0.58)	r = 0.03 (*p* = 0.95)	**r = 0.79** **(*p* = 0.03)**	r = −0.55 (*p* = 0.13)
**TRX1**	r = 0.28 (*p* = 0.46)		r = 0.40 (*p* = 0.29)	r = 0.02 (*p* = 0.98)	r = 0.35 (*p* = 0.36)	r = 0.32 (*p* = 0.41)	r = 0.08 (*p* = 0.84)	r = 0.15 (*p* = 0.71)	r = −0.03 (*p* = 0.95)	r = 0.28 (*p* = 0.46)
**sIL-5Rα**	r = −0.03 (*p* = 0.95)	r = 0.40 (*p* = 0.29)		**r = 0.81** **(*p* = 0.02)**	r = 0.50 (*p* = 0.18)	**r = 0.71** **(*p* = 0.04)**	r = 0.27 (*p* = 0.49)	**r = 0.88** **(*p* < 0.01)**	r = −0.40 (*p* = 0.29)	r = 0.27 (*p* = 0.49)
**MET**	r = −0.20 (*p* = 0.61)	r = 0.02 (*p* = 0.98)	**r = 0.81** **(*p* = 0.02)**		r = 0.57 (*p* = 0.12)	**r = 0.83** **(*p* = 0.02)**	r = −0.12 (*p* = 0.78)	**r = 0.93** **(*p* < 0.01)**	r = −0.18 (*p* = 0.64)	r = −0.22 (*p* = 0.58)
**IL-4**	r = 0.40 (*p* = 0.29)	r = 0.35 (*p* = 0.36)	r = 0.50 (*p* = 0.18)	r = 0.57 (*p* = 0.12)		**r = 0.91** **(*p* < 0.01)**	r = 0.50 (*p* = 0.18)	**r = 0.90** **(*p* < 0.01)**	r = 0.47 (*p* = 0.21)	r = −0.47 (*p* = 0.21)
**IL-5**	r = 0.17 (*p* = 0.68)	r = 0.32 (*p* = 0.41)	**r = 0.71** **(*p* = 0.04)**	**r = 0.83** **(*p* = 0.02)**	**r = 0.91** **(*p* < 0.01)**		r = 0.48 (*p* = 0.19)	**r = 0.83** **(*p* < 0.01)**	r = 0.15 (*p* = 0.71)	r = −0.32 (*p* = 0.41)
**IL-13**	r = 0.22 (*p* = 0.58)	r = 0.08 (*p* = 0.84)	r = 0.27 (*p* = 0.49)	r = −0.12 (*p* = 0.78)	r = 0.50 (*p* = 0.18)	r = 0.48 (*p* = 0.19)		r = 0.05 (*p* = 0.91)	r = 0.42 (*p* = 0.27)	r = −0.07 (*p* = 0.88)
**IL-25**	r = 0.03 (*p* = 0.95)	r = 0.15 (*p* = 0.71)	**r = 0.88** **(*p* < 0.01)**	**r = 0.93** **(*p* < 0.01)**	**r = 0.90** **(*p* < 0.01)**	**r = 0.83** **(*p* < 0.01)**	r = 0.05 (*p* = 0.91)		r = −0.12 (*p* = 0.78)	r = −0.28 (*p* = 0.46)
**Eotaxin-1**	**r = 0.79** **(*p* = 0.03)**	r = −0.03 (*p* = 0.95)	r = −0.40 (*p* = 0.29)	r = −0.18 (*p* = 0.64)	r = 0.47 (*p* = 0.21)	r = 0.15 (*p* = 0.71)	r = 0.42 (*p* = 0.27)	r = −0.12 (*p* = 0.78)		**r = −0.75** **(*p* = 0.03)**
**FcεRI**	r = −0.55 (*p* = 0.13)	r = 0.28 (*p* = 0.46)	r = 0.27 (*p* = 0.49)	r = −0.22 (*p* = 0.58)	r = −0.47 (*p* = 0.21)	r = −0.32 (*p* = 0.41)	r = −0.07 (*p* = 0.88)	r = −0.28 (*p* = 0.46)	**r = −0.75** **(*p* = 0.03)**	

TRX1—thioredoxin 1; ICTP—C-telopeptide of type I collagen; MET—tyrosine-protein kinase Met; sIL-5Rα—soluble interleukin 5 receptor subunit alpha; IL—interleukin; sFcεRI—soluble high-affinity IgE receptor; r—correlation coefficient. The statistically significant values are presented in bold. Statistical analysis: Spearman’s rank correlation.

## Data Availability

The original contributions presented in this study are included in the article. Further inquiries can be directed to the corresponding author.

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
