# Peer review of "Serum T2-High Inflammation Mediators in Eosinophilic COPD"

_biomolecules, 2024, doi:10.3390/biom14121648_

Round 1
Reviewer 1 Report
Comments and Suggestions for Authors
This article addresses a significant topic in respiratory medicine by exploring serum biomarkers of T2-high inflammation in eosinophilic COPD (eCOPD) and comparing them to severe non-allergic eosinophilic asthma (SNEA). The focus on personalized medicine is timely and relevant, especially in conditions like eCOPD that lack well-defined therapeutic targets. The study is well-structured, and the methodology is robust, with appropriate ethical and statistical considerations. However, several areas need refinement to enhance clarity, reproducibility, and the impact of the findings.
There are a few minor revisions to this article.
General Comments
Introduction: Please expand on the clinical implications of distinguishing eCOPD from SNEA beyond diagnosis. How could this influence treatment choices or health outcomes?
Methods:
· The study would benefit from a clearer justification for the small sample size
· Please provide more details about ELISA sensitivity and reproducibility. Were there any measures taken to control for batch effects or other technical variations?
· What were the criteria for defining "steroid-free" patients? No steroids ever or in the past month or?
Results:
· Clarify the clinical relevance of findings such as elevated MET levels in eCOPD. How do these findings align with or challenge existing literature?
· The discussion of eotaxin-1 levels is compelling but needs more context. Why might eotaxin-1 levels be lower in eCOPD compared to SNEA despite eosinophilic inflammation?
Discussion:
· The discussion around the T1 and T2 imbalance in eCOPD would benefit from integration with a more detailed exploration of therapeutic strategies.
· How could steroid use have influenced the biomarker levels observed?
· Maybe address smoking status
· The discussion regarding TRX1 could be clarified a bit, I think. As eCOPD had similar levels to HS and SNEA had decreased levels (why do you believe they were decreased?)
· Line 309: typo, please replace ”bendralizumab” with ”benralizumab”
· Line 312: ”It might affect biological medication action as organisms are adapted to regulate IL-5 activity.” Please reformulate, as this sentence is missing context.
References:
· The references are not consistently formatted – capitalization, abbreviation, italicization, etc
Author Response
Dear Reviewer,
We are thankful for Your comments and suggestions which have significantly contributed to the clarity of our manuscript. We hope that our corrections and additions meet Your expectations; however, if there are still any uncertainties or we made a correction not as You expected we can do it in the next revision stages. Please find an attachment with the full answers and links to the corrections.

Reviewer 2 Report
Comments and Suggestions for Authors
The research article "Serum T2-high Inflammation Mediators in Eosinophilic COPD 2" by A. Januskevicius et al, is well structured and the contents are clearly searched for a topic of great interest. A table with the comorbidities and therapy of the respiratory disease of each patient could be added. Also, can they have done the global Spirometry test?
Author Response
Dear Reviewer,
We are thankful for Your comments. We hope that our answers meet Your expectations; however, if there are still any uncertainties we can explain them in the next revision stages. Please find an attachment with the full answers and links to the edited text.

Reviewer 3 Report
Comments and Suggestions for Authors
Authors of the study attempt to assess potential biomarkers of eosinophilic phenotype of COPD. In my opinion heterogeneity of inflammation in COPD is an important and clinically relevant issue. Role of eosinophils in some patients with COPD, particularly during exacerbation, has been already demonstrated. However we still miss reliable biomarkers to confirm this phenotype or assess clinical course of the disease. In the era of biologicals studies related to mechanism of eosinophilic inflammation and its markers are particularly valuable as they may result in new treatment options.
Generally the manuscript in well written, methodology is clearly described and the results are presented transparently.
As far as methodology is concerned, it is unclear why the authors recruited patients with severe eosinophilic non-allergic asthma in order to compare this group with eCOPD patients not requiring steroids (i.e. very severe asthmatics and probably less severe patients with COPD). Such recruitment procedure could have affected the study results.
In Discussion section the authors need to refer more to earlier studies on biomarkers in COPD/asthma instead of details of mechanisms of activity of tested substances.
Author Response
Dear Reviewer,
We are thankful for Your comments and suggestions which have significantly contributed to the clarity of our manuscript. We hope that our corrections and additions meet Your expectations; however, if there are still any uncertainties or we made a correction not as You expected we can do it in the next revision stages. Please find an attachment with the full answers.

Reviewer 4 Report
Comments and Suggestions for Authors
This manuscript represents a study of type T2 high Inflammation mediators and markers in eosinophilic COPD and a comparison to the levels in healthy subjects as well as to patients with severe non-allergic eosinophilic asthma. The manuscript is generally well written. The authors discuss the limitations of the study in line 449-454 including the small number of study participants. The methods are straightforward, predominantly consisting of widely available analytical kits or physical tests. I have only minor comments for revisions.
1. Line 47: Perhaps the phrase, "can combine", should be replaced with, "are common to".
2. Lines 335-339: Perhaps the authors' comment, "TRX1 levels were significantly higher in eCOPD patients‘ serum compared to the severe asthma group; however, it did not differ from the HS group (Figure 2C). The asthma group were non-smokers while COPD patients were smokers or ex-smokers. It could be partly explained by this status as smokers have significantly higher serum thioredoxin concentrations," could be the explanation to their quandry. Asthma is largely a hyperresponse to either an irritant or immunological sensitization based stimulus. eCOPD has responses in common with certain types of asthma, as the authors show. However, a large portion of COPD patients are present or former smokers. The asthma patients in the study were not. Smokers expose their lungs to a large quantity of particulate, to which non-smokers are not as highly exposed. Among early smokers and seldom smokers, the predominant early response is the T1 type response which is essential to the attempt to clear the lungs of particulate. M1 type macrophages and dendritic cells play a large part in this. However, it has been noted that among heavy smokers, this response is suppressed, evident by greater susceptibility to pulmonary infections. While the M1/T1 responses may predominate in limited particulate exposures, a shift toward M2/T2 phenotypes has been observed to occur with chronic exposure, perhaps at points in which M1/T1 response is suppressed. There appears to be a continuum in proportion of both types of response among smokers depending on how heavily they smoke and the length of time they have smoked. The authors even mention this immunological continuum in lines 392-394, showing that they are aware, but perhaps have just not connected the dots. To summarize, this may help explain the quandry faced by the authors in this part of the discussion. The authors may decide whether to agree and edit their comments or rebut this suggestion for further thought.
3. Lines 379-383: SNEA is by definition non-atopic. However, eCOPD is associated with cytokines related to sensitization. Could that be part of the explanation for the immunological differences between SNEA and eCOPD discussed here?
Author Response

(The authors gave the same response as above.)
